# RNA binding protein IGF2BP2 expression is induced by stress in the heart and mediates dilated cardiomyopathy

Miriam Krumbein[1], Froma Oberman[1], Yuval Cinnamon[2], Mordechai Golomb[3], Dalit May[4,5,6], Gilad Vainer[7], Vitali Belzer[7], Karen Meir[7], Irina Fridman[1], Johannes Haybaeck[8,9], Gerhard Poelzl[10], Izhak Kehat[11], Ronen Beeri [12], Sonja M. Kessler[13,14 ✉] & Joel K. Yisraeli [1,14 ✉]

The IGF2BP family of RNA binding proteins consists of three paralogs that regulate intracellular RNA localization, RNA stability, and translational control. Although IGF2BP1 and 3 are oncofetal proteins, IGF2BP2 expression is maintained in many tissues, including the heart, into adulthood. IGF2BP2 is upregulated in cardiomyocytes during cardiac stress and remodeling and returns to normal levels in recovering hearts. We wondered whether IGF2BP2 might play an adaptive role during cardiac stress and recovery. Enhanced expression of an *IGF2BP2* transgene in a conditional, inducible mouse line leads to dilated cardiomyopathy (DCM) and death within 3-4 weeks in newborn or adult hearts. Downregulation of the transgene after 2 weeks, however, rescues these mice, with complete recovery by 12 weeks. Hearts overexpressing IGF2BP2 downregulate sarcomeric and mitochondrial proteins and have fragmented mitochondria and elongated, thinner sarcomeres. IGF2BP2 is also upregulated in DCM or myocardial infarction patients. These results suggest that IGF2BP2 may be an attractive target for therapeutic intervention in cardiomyopathies.

[1] Department of Developmental Biology and Cancer Research, Institute for Medical Research-Israel–Canada, Faculty of Medicine, Hebrew University of Jerusalem, Jerusalem, Israel. [2] Institute of Animal Science, Agricultural Research Organization, The Volcani Institute, Rishon Lezion, Israel. [3] The Heart Institute, Hadassah Medical Center, Jerusalem, Israel. [4] Faculty of Medicine, The Hebrew University of Jerusalem, Jerusalem, Israel. [5] Shaare Zedek Medical Center, Jerusalem, Israel. [6] Clalit Health Service, Jerusalem, Israel. [7] Department of Pathology, Hadassah Medical Center, Jerusalem, Israel. [8] Institut für Pathologie, Neuropathologie und Molekularpathologie, Medical University Innsbruck, Innsbruck, Austria. [9] Diagnostic and Research Center for Molecular Biomedicine, Institute of Pathology, Medical University of Graz, 8010 Graz, Austria. [10] Department of Cardiology and Angiology, Medical University Innsbruck, Innsbruck, Austria. [11] Department of Physiology and Biophysics, The Ruth and Bruce Rappaport Faculty of Medicine, Technion Israel Institute of Technology, Bat Galim, Haifa, Israel. [12] Department of Cardiology, Hadassah Medical Center, Jerusalem, Israel. [13] Experimental Pharmacology for Natural Sciences, Institute of Pharmacy, Martin Luther University Halle-Wittenberg, Halle, Germany. [14] These authors contributed equally: Sonja M. Kessler, Joel K. Yisraeli. ✉email: sonja.kessler@pharmazie.uni-halle.de; joel.yisraeli@mail.huji.ac.il

GF2BPs are a family of RNA-binding proteins composed of three paralogs, IGF2BP1, IGF2BP2, and IGF2BP3, that are highly conserved from fish and amphibians to mammals[1]. Enhanced CLIP analysis of human embryonic stem cells indicates that the IGF2BP paralogs (particularly IGF2BP1 and IGF2BP2) overlap extensively in their RNA targets[2]. Broadly expressed throughout embryonic development, IGF2BP1 and IGF2BP3 proteins are generally downregulated after birth, while IGF2BP2 expression is maintained in a number of adult tissues[3]. A wide variety of cancers show an upregulation of one or more of the IGF2BP proteins. The IGF2BP family of RNA binding proteins regulates RNA at many levels, including intracellular RNA localization, RNA stability, and translational control. The IGF2BPs are also reported to be readers of m6A RNA methylation, an epigenetic modification that can affect RNA stabilization and translation[4]. Given the apparently large number of potential targets, it is not surprising that IGF2BP proteins have been implicated in many cellular functions, including cell migration, proliferation, axon turning and guidance, maintenance of epithelial structures, and metabolic regulation[5–9].

Cardiac disease, one of the leading causes of death worldwide, affects millions of people yearly. RNA binding proteins (RBPs) have come under increased scrutiny for their role in cardiac health and disease[10–12]. Recently, a network of RBPs has been shown to associate with α-actinin in iPS-induced cardiomyocytes, helping to transport mitochondrial RNAs to the sarcomere and aiding in metabolic adaptations to stress[13]. Among these RBPs is IGF2BP2.

We report here that IGF2BP2 is normally expressed at low levels in postnatal and adult hearts, but that pressure overload or ischemia, both of which cause hypertrophy, cause an increase in IGF2BP2 levels. Furthermore, activating IGF2BP2 expression in either newborn or adult transgenic mice results in dilated cardiomyopathy leading to heart failure or cardiac dysfunction and precocious death within 3–4 weeks after induction of expression. Loss of heart function is progressive but reversible if exogenous IGF2BP2 expression is turned off in time. Adult hearts in which IGF2BP2 expression is activated for 2 weeks show a striking decrease in mitochondrial and sarcomeric proteins, as well as a shift in the expression of proteins associated with mitochondrial- to those associated with non-mitochondrial-based metabolism. Analysis of EM sections of these hearts reveals a preponderance of fragmented mitochondria and elongated, thinner sarcomeres, validating these findings. Consistent with the mouse data, elevated levels of IGF2BP2 are observed in human patients with dilated cardiomyopathies or after myocardial infarctions. These results suggest a model in which IGF2BP2 is upregulated upon various cardiac stresses and induces remodeling as part of an adaptive response. This response, however, comes at the price of impaired mitochondria and reduced sarcomere organization that eventually leads to heart failure (cardiac dysfunction) if IGF2BP2 levels are not reduced.

## Results

**IGF2BP2 is upregulated as a result of cardiac stress**. To better understand the role of IGF2BPs in the heart, we characterized the expression of all three IGF2BP paralogs in maturing hearts at both RNA and protein levels. As expected for oncofetal proteins, IGF2BP1 and IGF2BP3 RNA and protein levels decline rapidly after birth, with essentially baseline levels observed by 3 weeks of age. IGF2BP2 RNA expression, however, is maintained even in the adult heart, with a low but detectable level of protein still apparent at 6 weeks (Fig. 1a, b).

Cardiomyopathy, which leads to heart failure and death, can be induced by both ischemic and non-ischemic stress to the heart.

To investigate whether IGF2BP proteins might be activated during cardiac stress, we initially searched available RNA expression data from various cardiac stress models for expression of any of the IGF2BP paralog RNAs. The only paralog that was significantly upregulated in several mouse models of induced cardiac stress was IGF2BP2. Temporary ligation of the left anterior descending (LAD) coronary artery creates ischemia in the left ventricle, and this technique has been used as a model to induce myocardial infarction[14]. IGF2BP2 RNA was upregulated specifically in LAD ligated, but not sham-operated, mice at 24 and 48 h post occlusion, and only distal to the ligation (Supplementary Fig. 1a)[15]. Notably, IGF2BP2 expression returned to normal levels by 8 weeks, concomitant with recovery. A similar correlation between elevated IGF2BP2 expression and dilated cardiomyopathy and heart failure, followed by reduced IGF2BP2 expression upon resolution of heart failure, was observed in MerCreMer mice[16] (Supplementary Fig. 1b). IGF2BP2 was also activated by isoproterenol (Supplementary Fig. 1c[17]) and phospholamban-induced dilated cardiomyopathies[18].

We examined the response of IGF2BP2 to cardiac stress in two mouse models. In the first model, it had been reported that inducible expression of a soluble VEGF (sFlt) receptor in the heart of juvenile mouse pups causes chronic hypoxia, cardiac remodeling, and hibernation of cardiomyocytes[19]. When we analyzed the data that had been previously obtained from these mice, we found that IGF2BP2 RNA expression was upregulated over 4-fold as a result of this induced ischemia (Supplementary Fig. 1d). Silencing of the sFlt receptor, which rescued the phenotype, concomitantly downregulated IGF2BP2 expression. Furthermore, ChIP analysis of the IGF2BP2 gene revealed that H3K27 acetylation, a marker of transcriptional activation, varied with the activation and silencing of the sFlt receptor and mirrored mRNA expression levels of IGF2BP2 (Supplementary Fig. 1e). In a second model[20], transverse aortic constriction (TAC) generated pressure overload in the heart, a procedure that leads to cardiac hypertrophy and heart failure. We performed RT-PCR on RNA isolated from these hearts and observed an upregulation of IGF2BP2 RNA as compared to the sham-treated hearts (Fig. 1c). Upregulation of IGF2BP2 protein was also observed by immunofluorescent staining of sections from TAC versus sham-treated mice (Fig. 1d). Taken together, these experiments indicate that various types of stress that lead to cardiomyopathies and heart failure upregulate IGF2BP2 in the heart.

**IGF2BP2 upregulation in the heart leads to cardiac remodeling, fibrosis, heart failure and death**. Cardiac stress can have profound and pleiotropic effects on heart physiology. To test the direct effect of IGF2BP2 upregulation in the heart, we used a transgenic mouse system that enabled us to express human IGF2BP2 (hIGF2BP2) specifically in cardiac tissue in a conditional and reversible manner, in the absence of any induced stress. Transgenic mice expressing a tetracycline-regulated transactivator protein (tTA) from a heart-specific myosin heavy chain (MHC) α promoter were mated with mice harboring a transgene encoding an hIGF2BP2 protein under control of the tetracycline response element upstream of a minimal cytomegalovirus (TRE-CMVmin) promotor. In offspring possessing both transgenes, cardiac-specific expression of hIGF2BP2 can be regulated by tetracycline through the addition of, or withdrawal from, the drinking water (off or on, respectively; Supplementary Fig. 2a, b). Transgene expression in cardiac tissue from these mice was confirmed by histological staining (Supplementary Fig. 2c), and the level of expression of the hIGF2BP2 transgene was ~2.5-fold above that of the endogenous mouse IGF2BP2, as determined

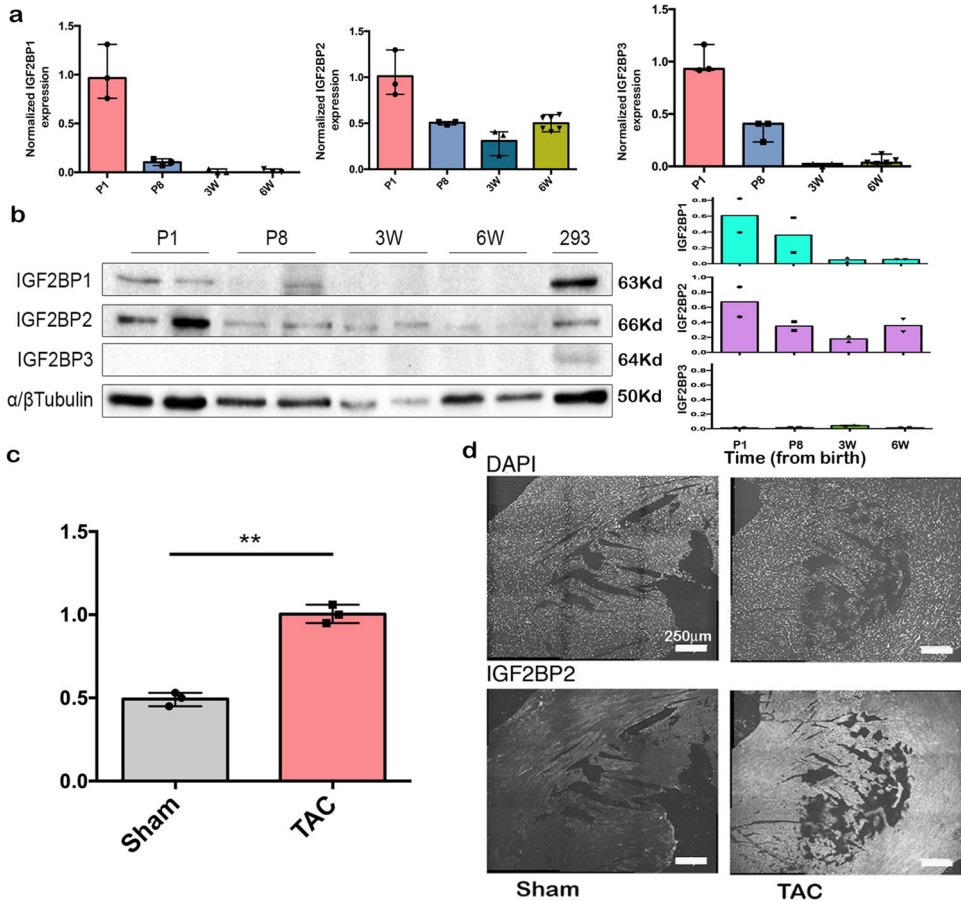

**Fig. 1 Expression of IGF2BP paralogs and cardiac stress models. a** Relative *IGF2BP2* RNA expression was measured by RT-PCR in normal hearts at postnatal day 1 (P1), postnatal day 8 (P8), 3 weeks (3 W), and 6 weeks (6 W) of age. Each bar represents three biological repeats ±SEM). **b** Western blot analysis of IGF2BPs at the indicated time points. α/β Tubulin was used as a loading control. On the right, is the quantification of the normalized results. Two biological repeats are shown for each time point ±SEM. Each protein was probed on a different blot. **c** mRNA was extracted from the hearts of sham or TAC mice 10 weeks after the procedure and assayed by RT-PCR for *IGF2BP2* expression (*n* = 3; *p* = 0.0015). **d** IGF2BP2 expression was detected by immunofluorescence (bottom row) on heart sections from sham-operated or TAC mice. Nuclei are stained with DAPI (top row). Scale bar is 250 μm. Statistical significance was determined by a two-tailed Student's *t*-test. **\*\****p* < 0.01.

by RT-PCR using RNA isolated from primary cardiomyocytes grown from mice induced for 5 days (Supplementary Fig. 2d).

When tetracycline is not administered to pregnant mice containing both transgenes, *hIGF2BP2* expression is activated at the time that the MHCα gene is first turned on, around day E8.5 of development[21]. Pups appear normal at birth but all of them die very early, even before weaning, beginning at 21 days (Fig. 2a). When sacrificed at 12 days of age, these mice have enlarged hearts (Fig. 2b, c). In histological sections stained for H&E, it is clearly apparent that the *hIGF2BP2*-expressing hearts are much larger and remodeled, and that their endo and sub-endocardium are less dense than compared to those of control hearts (Fig. 2d).

To determine whether upregulation of *IGF2BP2* can affect not only perinatal but also adult hearts, mice were maintained on tetracycline for 8–10 weeks (*IGF2BP2* off) before the antibiotic was removed from their drinking water. Induction of *hIGF2BP2* at this point also leads to heart failure and death in all the mice, although the process takes slightly longer in females (26–32 days) as opposed to males (21–26 days; Fig. 2e). As observed in the perinatally induced mice, the hearts were enlarged and remodeled, and these modifications progressed gradually from 10 to 28 days post-*hIGF2BP2* induction (Fig. 2f, g). In histological slides prepared from these hearts at the different time points, the changes observed at 10 days post-induction are very subtle, with

slight disarray and mismatch of cell size of the cardiomyocytes, mainly in the papillary muscle and the sub-endocardium. With time, these differences become more pronounced, and fibrosis increases, also localizing to the sub-endocardium close to the lumen of the left ventricle (Fig. 2h and Supplementary Fig. 2e, f).

Heart function was monitored in these mice by echocardiography (Fig. 2i–l). Prior to induction of the *IGF2BP2* gene, the hearts functioned normally in each of the three cohorts (MHC-tTA/IGF2BP2, MHC-tTA, and WT), indicating that there was no leakiness of *hIGF2BP2* expression. At 10 days post-induction, although the cardiac stress/DCM marker α-actin 1 is already significantly upregulated at the transcriptional level (Fig. 2o), the heart still appears to function normally, with no apparent differences observed in ejection fraction or fractional shortening. At 20 days post-induction, however, mice expressing *hIGF2BP2* exhibit a clear DCM phenotype[22], with ejection fraction under 40%, fractional shortening under 20%, LV end-diastolic diameter enlarged by more than 120%, and with 4 out of 6 of the mice showing significant dyssynchrony (Fig. 2i–l), as often associated with DCM[23]. Furthermore, by this point after induction, the stress markers ANP, BNP, and α-actin 1 are all notably elevated (Fig. 2m–o). No arrhythmias were detected in any of the *hIGF2BP2*-expressing mice (Fig. 2i). These results argue that upregulation of *IGF2BP2* expression rapidly induces stress in

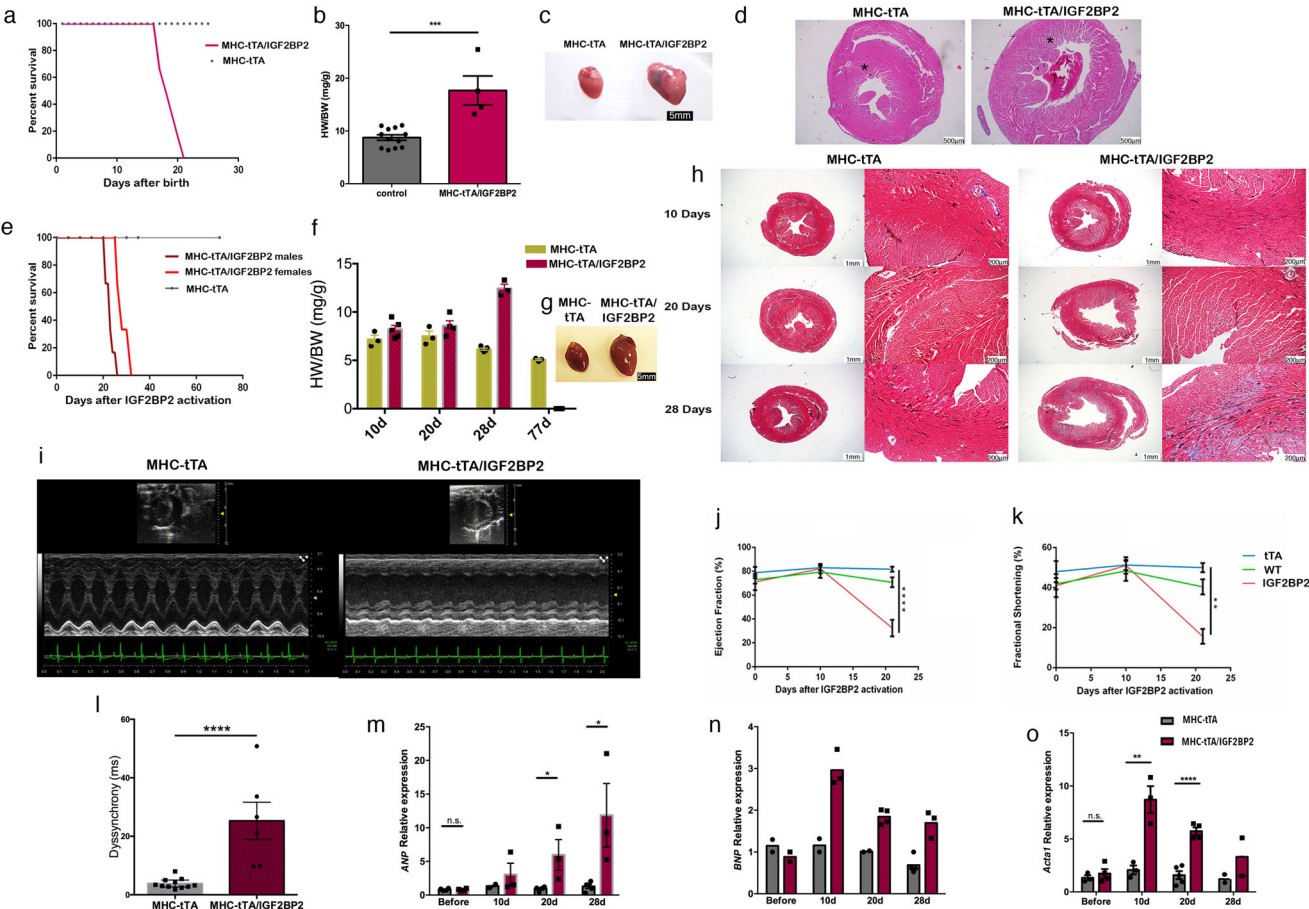

**Fig. 2 Activation of the conditional IGF2BP2 transgene in mouse hearts. a** Kaplan–Meier survival curve of the control (MHC-tTA, $n = 6$) and IGF2BP2-expressing transgenic (MHC-tTA/IGF2BP2, $n = 6$) mice. Pregnant mothers were fed water without tetracycline (i.e., activation of the transgene occurred from E8.5). **b** Scatter plot/bar graph of the ratio of heart weight to body weight in control and MHC-tTA/IGF2BP2 mice. **c** Representative pictures of an MHC-tTA (left) heart and an MHC-tTA/IGF2BP2 heart (right) ±SEM. **d** H&E staining of a cross-section of an MHC-tTA (left) and MHC-tTA/IGF2BP2 heart (right). Scale bar is 500 μm. Asterisks indicate the sub-endocardium. **e** Kaplan–Meier survival curve of 8–10 weeks old transgenic mice (MHC-tTA/IGF2BP2, males $n = 6$, females $n = 6$; MHC-tTA, males $n = 6$, females $n = 5$). Tetracycline was withdrawn from drinking water on day 0. **f** Bar graph of the ratio of heart weight to body weight (HW/BW mg/g ± SEM) in adult MHC-tTA mice and MHC-tTA/IGF2BP2 mice at different time points after transgene induction (10d: MHC-tTA, $n = 3$; MHC-tTA/IGF2BP2, $n = 5$; 20d: MHC-tTA, $n = 3$; MHC-tTA/IGF2BP2, $n = 4$; 28d: MHC-tTA, $n = 5$; MHC-tTA/IGF2BP2, $n = 3$). **g** A representative picture of an MHC-tTA heart (left) next to an MHC-tTA/IGF2BP2 heart (right) at endpoint. **h** Trichrome staining of MHC-tTA (left) and MHC-tTA/IGF2BP2 (right) hearts at the indicated time points. Scale bar is 1 mm and 100 μm in the higher magnification (right panels). **i** Representative pictures of the echocardiogram at 21 days post activation. **j** and **k** Echocardiograms were performed on WT, MHC-tTA, and MHC-tTA/IGF2BP2 mice before, 10 days post, and 21 days post activation of the transgene ($n = 6$ in each group). Analyses are shown of **j** Ejection fraction percentage and **k** Fractional shortening percentage. Statistical significance was determined by one-way analysis of variance (ANOVA) followed by Dunnett's post hoc test. **l** Dyssynchrony was measured in M-mode as the delay in the onset of positive peak systolic velocity 21 days post-activation. Each dot represents one mouse. **m** and **o** RNA was extracted from MHC-tTA and MHC-tTA/IGF2BP2 hearts at the indicated time points and analyzed by RT-PCR for expression of ANP (**m**), BNP (**n**), and ACTA1 (**o**). Values show the results from 2 to 5 biological repeats ±SEM (3 technical repeats/sample). Statistical significance in the bar graphs was determined by a two-tailed Student's $t$-test. $*p < 0.05$, $**p < 0.01$, $***p < 0.001$, $****p < 0.0001$.

cardiomyocytes and is sufficient to cause DCM progressing to heart failure and death in both perinatal and adult hearts within 3–4 weeks.

**IGF2BP2-induced cardiomyopathy is reversible**. In both the LAD occlusion model of MI (Supplementary Fig. 1a) and the MerCreMer model (Supplementary Fig. 1b), *IGF2BP2* appears to return to nearly normal levels several weeks after the initial stress, and this drop correlates with the rescue of the cardiac myopathy phenotype. These findings prompted us to test whether cessation of *hIGF2BP2* expression after induction of cardiac remodeling could rescue mice from progressing to complete heart failure. We employed two groups of transgenic mice. In the first group, *hIGF2BP2* expression was induced for 2 weeks in 11 adult mice

(8–10 weeks old) and then switched off (Fig. 3a). No loss of heart function was observed (Fig. 3b, c), and there was a 100% survival rate. This was in striking contrast to the 0% survival rate at 32 days in the mice that continued to express *hIGF2BP2* beyond 2 weeks (Fig. 2e). Three of the surviving mice in the first group were sacrificed 4 weeks later, and their hearts were found to be slightly enlarged, indicating that some remodeling had occurred as a result of the 2-week induction (Fig. 3d). In these hearts, wheat germ agglutinin staining of the endocardium adjacent to the papillary muscle confirmed that the cardiomyocytes were hypertrophic (Fig. 3e). The 8 remaining mice survived until they were sacrificed, 12 weeks after the gene had been turned off. Heart function was not affected throughout the entire post-*IGF2BP2* expression period (Fig. 3b, c), and their hearts returned to normal

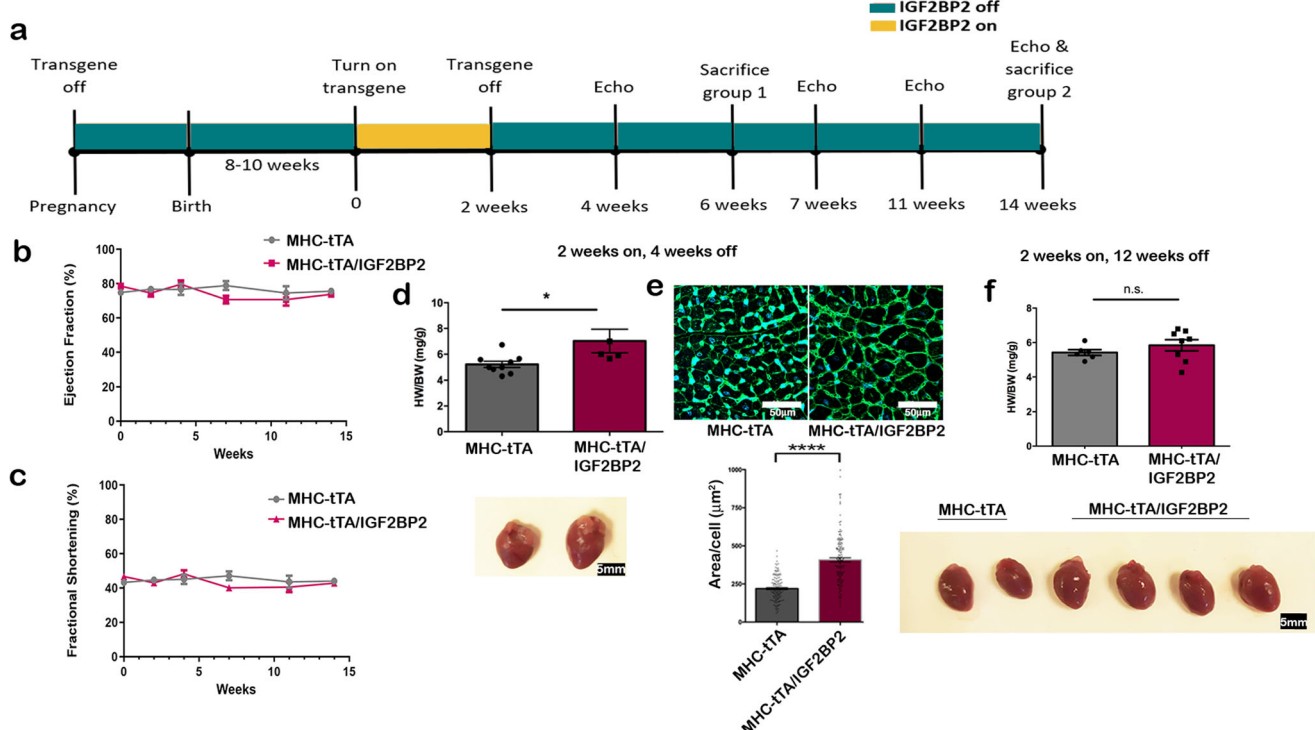

**Fig. 3 Rescue experiment with full recovery. a** Timeline of experimental design for the experiments described in (**b**–**f**): transgene on for 2 weeks and off for 12 weeks in adult mice. **b** and **c** Echocardiograms were performed on all the mice over the course of the experiment and analyzed for Ejection Fraction percentage (**b**) and fractional shortening percentage (**c**). **d** Scatter plot/bar graph of the ratio of heart weight to body weight (HW/BW) (mg/g) ±SEM in mice after 2 weeks of induction and 4 weeks after turning off the transgene. Below the graph are representative pictures of an MHC-tTA heart (left) next to an MHC-tTA/IGF2BP2 heart (right) ($p = 0.0299$) (MHC-tTA, $n = 9$; MHC-tTA/IGF2BP2, n = 5). **e** Wheat germ agglutinin staining of endocardial tissue adjacent to the papillary muscle from the hearts described in (**d**). Quantification of the area of 150 cardiomyocytes from each cohort is shown below. **f** Scatter plot/bar graph of the ratio of heart weight to body weight (HW/BW) (mg/g) ±SEM in mice after 2 weeks of induction and 12 weeks after turning off the transgene. Below the graph are representative pictures of two MHC-tTA (left) and four MHC -tTA/IGF2BP2 hearts (right) (MHC-tTA $n = 5$, MHC-tTA/IGF2BP2, $n = 8$). Statistical significance was determined by a two-tailed Student's *t*-test. *$p < 0.05$, ****$p < 0.0001$.

size (Fig. 3f). These findings indicate that the effects of *IGF2BP2* expression on cardiac remodeling and function are potentially reversible.

A second group of 5 mice was induced for *hIGF2BP2* expression for 16 days (Fig. 4a). As in the first group, at the end of the induction there was no loss of heart function (Fig. 4b, c). Nevertheless, the outcome in this group was variable. Two mice (I and II) died within 2 weeks of turning off *hIGF2BP2*. The heart from mouse II had extensive dilation of both ventricles and loss of muscle wall thickness (Fig. 4f). Although the remaining three survived until the end of the experiment, at 12 weeks post-*hIGF2BP2* expression, one of the mice (mouse III) had severely impaired heart function (ejection fraction and fractional shortening; Fig. 4b, c) and a significantly large heart with extensive fibrosis throughout all of the left ventricle wall (Fig. 4e, f). The remaining two mice (IV and V) displayed some impairment in heart function that almost reverted back to normal by the end of the experiment, and their hearts were of normal size and with minimal fibrosis (Fig. 4d, f). The variability among this set of mice suggests that at some time slightly after 14 days of *hIGF2BP2* expression, remodeling of the heart progresses beyond a point of no return when the damage to the heart cannot be overcome, even if the gene is turned off.

**Major protein changes are found in sarcomeres and mito-chondria.** To elucidate the underlying mechanisms through which IGF2BP2 helps mediate dilated cardiomyopathies, we prepared protein extracts from hearts expressing either the

*IGF2BP2* transgene or the Tet transactivator alone for 15 days and analyzed them by mass spectrometry (MS; Supplementary Fig. 3a, b; Supplementary Data 1). Unsupervised hierarchical clustering identified two distinct groups corresponding to the tTA control and *IGF2BP2* transgene-expressing mice (Supplementary Fig. 3c; Supplementary Data 1). Supervised hierarchical clustering of differentially expressed genes showed a significant upregulation of 587 proteins and a significant downregulation of 440 proteins (FDR = 0.05, $S = 0.1$) (Fig. 5a). Among the proteins that were significantly affected by *hIGF2BP2* expression, 71 of them (41 upregulated and 30 downregulated) are associated with con-tractile proteins, as defined in Gene Ontology—Cell Components (GO:0043292). Another 407 of the affected proteins (98 upre-gulated and 309 downregulated) are correlated with Mitochon-drion Cell Components (GO:0005739). A few of these were validated by western blot analysis (Supplementary Fig. 3d, e).

At least 115 genes have been associated with various familial DCM syndromes[22,24–27]. Of these, 28 genes are associated with sarcomeres in general or Z discs in particular. Strikingly, 18 of these sarcomeric genes encode proteins significantly down-regulated by *hIGF2BP2* expression in the transgenic mice, and 10 of these are considered definitive for the DCM phenotype (Table 1). These results correlate nicely with the pathology and physiology of the transgenic hearts, further supporting the evidence that the expression of IGF2BP2 in the heart causes DCM.

A GO analysis of the biological processes and cellular components of proteins significantly downregulated by *IGF2BP2*

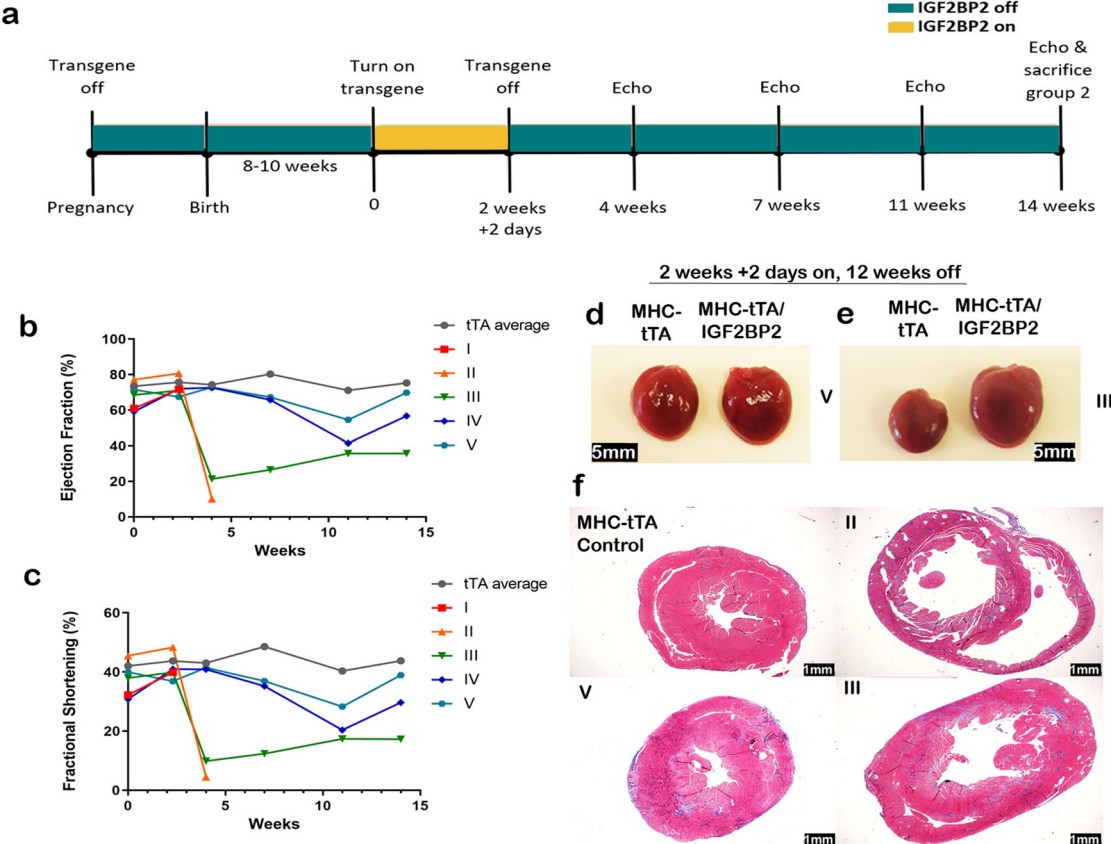

**Fig. 4 Rescue experiment with only partial recovery. a** Timeline of experimental design for the experiments described in (**b–f**): transgene on for 2 weeks and 2 days and off for 12 weeks in adult mice. Echocardiograms were performed on all the mice over the course of the experiment and analyzed for Ejection Fraction percentage (**b**) and fractional shortening percentage (**c**). Three MHC-tTA controls were averaged together and are graphed in gray. The five MHC-tTA/IGF2BP2 mice are graphed individually and numbered I–V. **d**, **e** Picture of hearts at the end of the 2 weeks and 2 days transgene induction followed by 12 weeks off: **d** MHC-tTA control heart (left) and an MHC-tTA/IGF2BP2 heart (right) with normal EF% (mouse no. V) and **e** MHC-tTA control heart (left) and an MHC-tTA/IGF2BP2 heart (right) with impaired EF% (mouse no. III). **f** Trichrome staining of an MHC-tTA control and the different outcomes of the MHC-tTA/IGF2BP2 mice at the end of the experiment. Scale bar is 1 mm.

expression (Fig. 5b and Supplementary Data 1) reveals a strong correlation with mitochondrial (309/440 = 70%, FDR = $2.04 \times 10^{-216}$) and metabolic processes (315/440 = 72%, FDR = $4.54 \times 10^{-55}$). The same analysis of upregulated proteins also shows a strong correlation with metabolic processes, albeit different proteins (Fig. 5c and Supplementary Data 1). However, while a similar number and percentage of the upregulated proteins are associated with metabolic processes (359/587 = 61%, FDR = $2.82 \times 10^{-41}$), only 98 of the upregulated proteins are associated with mitochondria (17%). To gain insight into why some of the proteins associated with metabolic processes are downregulated while others are upregulated by *IGF2BP2* expression, we correlated mitochondrial association with metabolic processes for each group. As seen in Supplementary Fig. 4, over 75% of the downregulated metabolic processes proteins are mitochondrial, as opposed to only 20% of the upregulated proteins. This highly significant correlation ($\chi^2 < 0.0001$) indicates that *IGF2BP2* expression in the heart causes mitochondrial proteins, and, as a result, mitochondrial metabolism, to be greatly diminished; non-mitochondrial metabolic pathways, in contrast, are upregulated, apparently in an attempt to compensate for the heart's metabolic requirements.

To better understand how IGF2BP2 might be influencing protein expression, we isolated RNA from the same control or *IGF2BP2* overexpression hearts used for the proteomics analysis and used RT PCR to compare the gene expression of several

RNAs whose encoded proteins were significantly downregulated in the mass spectrometry analysis. Of the seven RNAs analyzed, five of them, three mitochondrial (*Cox4I1*, *Cox5a*, and *OPA1*) and two sarcomeric (*Troponin I3* and *Myosin light chain 2*) demonstrated a significant reduction in RNA levels (Supplementary Fig. 5). Two other sarcomeric genes (*MYBPC3* and *TTN*) showed no significant change in RNA levels as a result of *IGF2BP2* expression. These results raise the possibility that IGF2BP2 may be acting in different ways to post-transcriptionally regulate RNAs in the heart (see the "Discussion" section).

To analyze the effect of IGF2BP2 on sarcomere structure at the cellular level, *IGF2BP2* expression was activated in primary cardiomyocytes for 7 days. Cells were stained with anti-IGF2BP2, anti-MF20 (a pan myosin heavy chain antibody), and DAPI (Supplementary Fig. 6a), and images were analyzed using an algorithm in ImageJ which highlights the direction of polarized pixels, based on coherency mapping[28]. As seen in Supplementary Fig. 6b, while control cardiomyocytes reveal a bright MF20 staining upon coherency mapping, indicating a more global orientation of myofibrils within the cell, IGF2BP2-expressing cardiomyocytes lose organization of pixels in the MF20 staining and exhibit a duller, more spotted appearance.

These results suggested that changes in cardiomyocyte structure upon IGF2BP2 induction might also be apparent at the ultrastructural level. Transmission electron microscopy (EM) images were acquired from the LV of adult hearts expressing

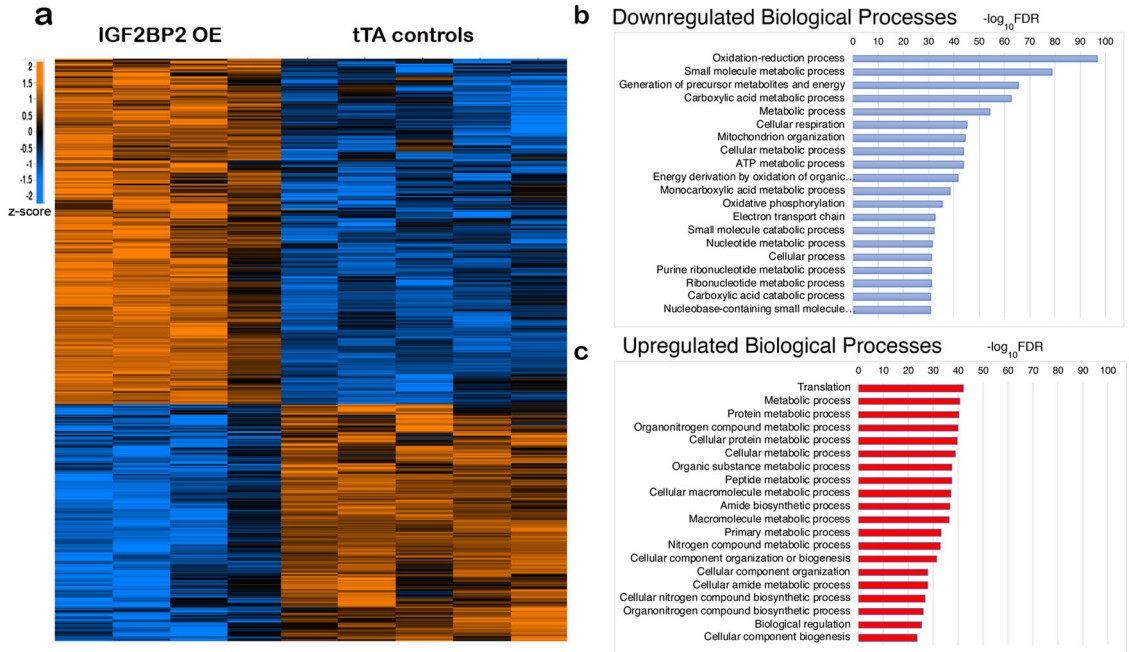

**Fig. 5 Proteomics of uninduced and induced adult hearts.** Protein extracts were prepared from MHC-tTA adult mice hearts (five samples, tTA controls) or from same-age siblings MHC-tTA/IGF2BP2 mice in which IGF2BP2 expression was activated for 15 days (four samples, IGF2BP2 OE) and analyzed by mass spectrometry. **a** Supervised hierarchical clustering of all 9 samples of differentially expressed proteins (with Z score, FDR = 0.05, S = 0.1; orange, upregulated; blue, downregulated). Top 20 Biological processes in GO analysis were ranked by $-\log_{10}$FDR (false discovery rate) of downregulated (**b**) and upregulated (**c**) proteins.

either *MHC-tTA* alone or *MHC-tTA/IGF2BP2*, 2 weeks after activating transgene expression (Fig. 6a). In accordance with the proteomics and myofibril staining data, the sarcomeres of the tTA-IGF2BP2 heart showed an altered structure, with the distance between the Z lines elongated (sarcomere length) and the width of the myofibrils shortened (Fig. 6b, c), a phenotype observed in dilated cardiomyopathy (DCM)[29,30].

The proteomics data predicted that *IGF2BP2* expression should have a strong effect on the mitochondria in transgenic hearts. Indeed, upon *IGF2BP2* overexpression, a disordered alignment of mitochondria is observed in EM grids, along with alterations in shape and size (Fig. 6a, d). Quantitative analysis shows that the number of mitochondria is 19% higher in the *IGF2BP2*-induced hearts (Fig. 6e), although the average size was reduced by 23% (Fig. 6f). Given that the total mitochondrial area per field is the same in both the control and overexpression hearts (Fig. 6g), these results indicate that the mitochondria are fragmented and not undergoing fusion properly when *IGF2BP2* expression is activated. In a rat coronary artery constriction model, heart failure was shown to cause a similar phenotype of increased numbers of fragmented, smaller mitochondria[31]. OPA1 is a mitochondrial inner membrane protein that helps regulate the fusion of mitochondria and is downregulated by ischemia and heart failure. Knockdown of OPA1 causes fragmentation of mitochondria in cultured cardiomyocytes[31], and, in mice, leads to heart failure[32]. Notably, OPA1 is one of the mitochondrial proteins downregulated after IGF2BP2 activation, at the level of both its RNA (Supplementary Fig. 5) and protein (Supplementary Data 1). It is interesting to note that OPA1 is also an essential protein for initiating mtDNA replication and maintaining mtDNA integrity[33]. The abundance of a mitochondrial DNA sequence was compared to that of a nuclear genomic DNA sequence in DNA extracted from the same hearts used for the MS analysis. We found that there was almost 40% less mtDNA in the *IGF2BP2*-induced hearts than in the MHC-tTA controls (Fig. 6h).

Taken together, these results confirm that overexpression of *IGF2BP2* in the heart causes a profound impairment of both sarcomere and mitochondrial morphology.

**IGF2BP2 expression in patients after MI or with DCM**. Having observed that *hIGF2BP2* expression in mouse hearts is sufficient to cause dilated cardiomyopathy and that *IGF2BP2* RNA is up-regulated in hearts that have undergone cardiac stress, we wondered whether IGF2BP2 protein expression is elevated in human pathologies. Heart tissue from patients with various cardiac diseases was stained for IGF2BP2 protein (Fig. 7). While IGF2BP2 expression was not observed in control or hypertrophic heart samples, IGF2BP2 expression was indeed detected in both myocardial infarction and DCM hearts, with expression significantly higher in DCM as compared to MI sections (Fig. 7e). These findings in human patients correlate well with the results from the mouse experiments, showing that IGF2BP2 is up-regulated upon cardiac stress and helps to remodel the heart.

## Discussion

Here we demonstrate the involvement of IGF2BP2 in response to cardiac stress and in the development of cardiomyopathies. IGF2BP2 is normally expressed in the heart during embryogenesis, and its expression is maintained at a low level postnatally into adult life. Expression of transgenic *IGF2BP2* in the mouse heart, either from birth or in the adult, causes an enlarged, remodeled, malfunctioning heart that leads to death within a period of ~3–4 weeks. Strikingly, *IGF2BP2* is upregulated not only in mouse models of cardiac ischemia and pressure overload but also in human patients with dilated cardiomyopathy and after myocardial infarction (Fig. 7). In several mouse models of induced cardiac stress, rescue of the heart and recovery of normal morphology is correlated with downregulation of *IGF2BP2*, raising the possibility that upregulation of IGF2BP2 under stress

**Table 1 Genes associated with DCM and their regulation by IGF2BP2 expression.**

| Gene code | Gene name | Cellular location | Regulation by IGFBP2 | Reference |
|---|---|---|---|---|
| LMNA | Lamin A/C | Nuclear envelope | | Pereira, 2020 |
| RBM20 | RNA-binding protein 20 | Nuclear | | Pereira, 2020 |
| TTN | Titin | Sarcomere | Down | Pereira, 2020 |
| MYH7 | Myosin-7 | Sarcomere | | Pereira, 2020 |
| MYH6 | Myosin-6 | Sarcomere | Down | Pereira, 2020 |
| TNNT2 | Troponin T, cardiac muscle | Sarcomere | Down | Pereira, 2020 |
| MYBPC3 | Myosin-binding protein C, cardiac-type | Sarcomere | Down | Pereira, 2020 |
| TNNI3 | Troponin I, cardiac muscle | Sarcomere | Down | Pereira, 2020 |
| TPM1 | Tropomyosin alpha-1 chain | Sarcomere | Down | Pereira, 2020 |
| MYL2 | Myosin regulatory light chain 2 | Sarcomere | Down | Pereira, 2020 |
| MYL3 | Myosin light chain 3 | Sarcomere | Down | Pereira, 2020 |
| ACTC1 | Alpha-cardiac actin | Sarcomere | | Pereira, 2020 |
| ACTN2 | Alpha-actinin 2 | Z-disc | | Pereira, 2020 |
| ANKRD1 | Ankyrin repeat domain-containing protein 1 | Z-disc | | Pereira, 2020 |
| CSRP3 | Cysteine and glycine-rich protein 3 | Z-disc | Down | Pereira, 2020 |
| LDB3 | LIM domain-binding protein 3 | Z-disc | Down | Pereira, 2020 |
| MYOZ2 | Myozenin-2 | Z-disc | Down | Pereira, 2020 |
| TCAP | Telethonin | Z-disc | Down | Pereira, 2020 |
| VCL | Vinculin | Z-disc | Up | Pereira, 2020 |
| MYPN | myopalladin | Z-disc | | Pereira, 2020 |
| DES | Desmin | Intermediate filament | | Pereira, 2020 |
| FLNC | Filamin-C | Intermediate filament | Up | Pereira, 2020 |
| SYNM | Synemin | Intermediate filament | | Maggi, 2021 |
| BAG3 | Co-chaparone 3 | Intracellular chaperone | | Pereira, 2020 |
| DSP | Desmoplakin | Intercalated disc | Up | Pereira, 2020 |
| PKP2 | Plakophilin 2 | Intercalated disc | | Pereira, 2020 |
| TAZ | Tafazzin | Mitochondrial membrane | | Pereira, 2020 |
| DTNA | Dystrobrevin-alpha | Plasma membrane | | Pereira, 2020 |
| SCN5A | Alpha-subunit of the cardiac sodium channel | Plasma membrane | | Pereira, 2020 |
| JPH2 | Junctophilin-2 | Sarcoplasmic reticulum, plasm membrane | Down | Wehrens,2022 |
| PLN | Phospholamban | Sarcoplasmic reticulum | | Sillje, 2021 |
| CRYAB | Alpha-crystallin B chain | Nucleus, cytoplasm, membrane | Down | Sillje, 2021 |

The table shows genes in which mutations have been associated with a DCM phenotype. The column labeled "Regulation by IGF2BP2" indicates whether the protein is upregulated, downregulated, or not significantly changed (blank) in hearts induced to express hIGF2BP2 (based on Supplementary Data 1).

conditions may initially be a compensatory response that later needs to be restricted to enable full recovery.

This hypothesis would predict that turning off *IGF2BP2* expression in a timely fashion could reverse the DCM phenotype it induced. Using a conditional mouse model, *IGF2BP2* was upregulated in adult hearts for 2 weeks and then turned off. Although increased expression of *IGF2BP2* for 2 weeks causes a DCM phenotype with the associated morphological, and structural changes in the myocardium and in gene expression, these changes were not severe enough to impair heart function. In addition, all these mice ultimately recovered, and their hearts returned to normal size. Increasing the overexpression period by only two days (16 as opposed to 14 days) caused a significant change in the phenotype of the mice: two mice died within 2 weeks of turning off *hIGF2BP2*, one mouse survived until the end of the experiment (12 weeks) but had an enlarged, malfunctioning heart, and only two mice were healthy 3 months post-abrogation of *hIGF2BP2*. These results indicate that, in mice, IGF2BP2 upregulation triggers a spiral into cardiac disease that is initially reversible but becomes independent of IGF2BP2 expression after 2 weeks.

The mechanisms by which overexpression of IGF2BP2 leads to DCM and death appear to be multiple and complex. In our proteomics analysis, over 300 mitochondrial proteins were significantly reduced by *hIGF2BP2* expression, representing by far the highest-ranking Cellular Component GO group influenced by IGF2BP2. The short, fragmented mitochondria phenotype observed in the EM sections of the *hIGF2BP2* expressing hearts is fully consistent with the broad downregulation of mitochondrial proteins and appears very similar to the mitochondria seen in ischemic heart failure[31]. Mitochondria constitute approximately 40% of the cytoplasm of an adult cardiomyocyte[34], and mitochondrial dysfunction has long been known to be associated with the development of heart disease[35]. Because many of the downregulated proteins are also classified as metabolic, such a large-scale assault on mitochondrial proteins can explain why these hearts proceed to heart failure. Accordingly, energy must be generated in these mice via alternative pathways, and we in fact observe this compensation in the upregulation of many other non-mitochondrial, metabolic proteins. A very similar phenotype, involving downregulation of aerobic mitochondrial metabolism genes and an upregulation of glycolytic glucose metabolism genes, is observed in mice induced to develop DCM by a mutation in phospholamban[36]. Similarly, genetic mitochondrial diseases, where the function of the mitochondria is compromised by mutations in mitochondrial proteins, are associated with cardiac abnormalities as well. Electrocardiographic and/or echocardiographic abnormalities are found in about 30% of adults with mitochondrial disease[37–40]. Although no arrhythmias were observed after inducing *hIGF2BP2* expression for 21 days in adult mice, four out of six of the mice showed dyssynchrony (Fig. 2l). Improving mitochondrial function is considered an emerging approach for treating heart failure[41].

IGF2BP2 has been shown to associate with αActinin2 protein in human cardiomyocytes induced from pluripotent stem cells, where it helps mediate intracellular localization of RNAs

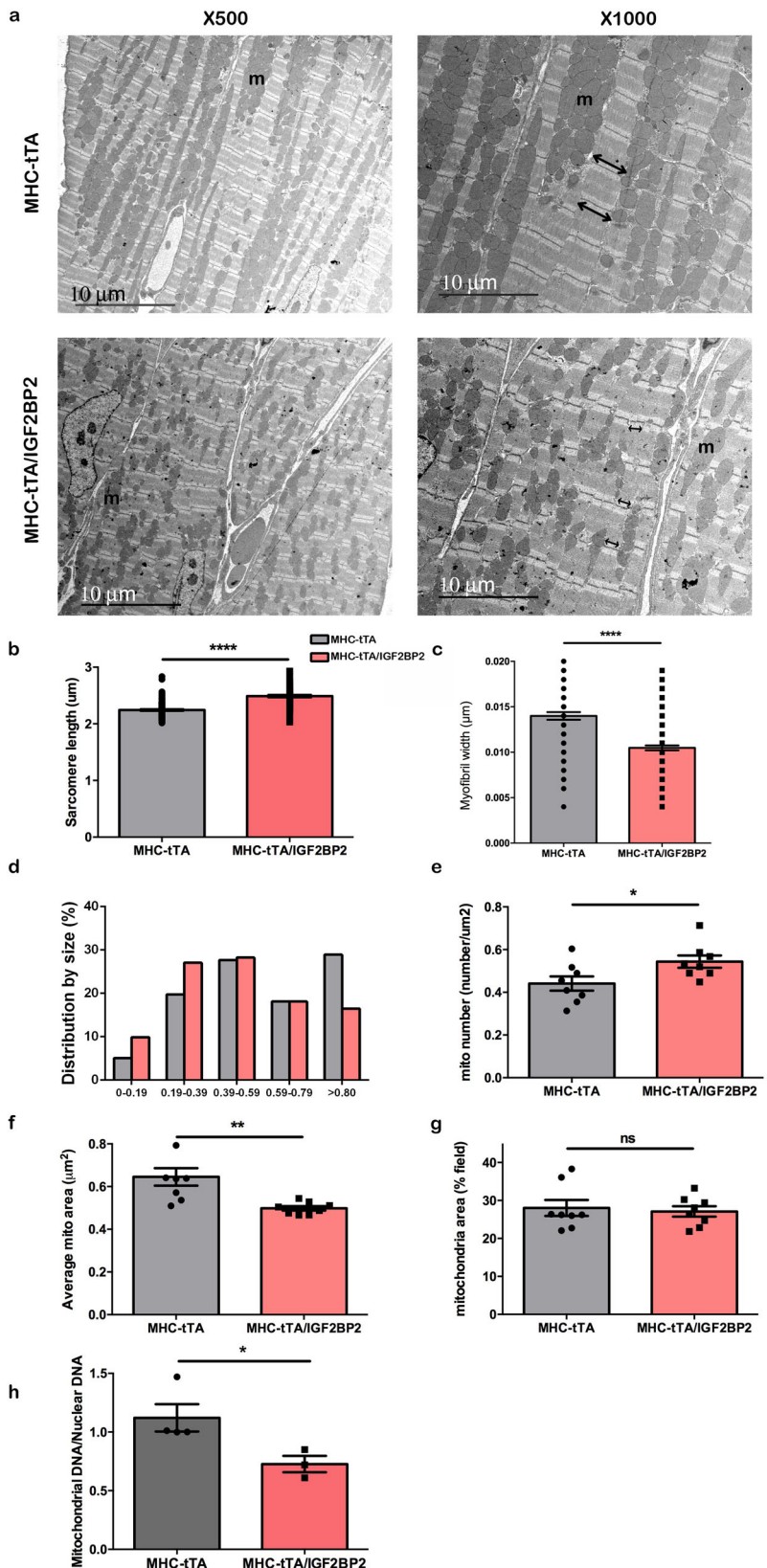

encoding mitochondrial proteins to the vicinity of the sarcomere[13]. The mechanism by which induced IGF2BP2 expression leads to downregulation of mitochondrial proteins is still unclear. IGF2BP2 has been shown to repress translation in several different settings, including translation of mitochondrial *UCP* RNA in brown fat[42]. Strikingly, these authors observed that over 40% of the RNAs associated with IGF2BP2 (as seen by immunoprecipitation) are mitochondrial. *OPA1* mRNA, which encodes a mitochondrial inner membrane fusion protein that causes mitochondrial fragmentation when downregulated, is one such target of IGF2BP2[2]. When IGF2BP2 is upregulated, we indeed observe that OPA1 protein levels are reduced

**Fig. 6 EM images of the myocardium from transgenic mice. a** Representative EM image of the LV myocardium of mice. Left panels, magnification was ×500, and right panels, ×1000. "m" indicates mitochondria. Note the absence of the H zone in both the MHC-tTA and MHC-tTA/IGF2BP2 hearts, indicating that both are in contraction. The double-headed arrows indicate the width of the myofibrils in the myocardium. Activation of IGF2BP2 expression leads to increased sarcomere length (**b**; $p < 0.001$) and decreased myofibril width (**c**; $p < 0.001$). IGF2BP2 activation also causes fractionalization of mitochondria, leading to an accumulation of smaller mitochondria (**d**) with a concomitant increase in their number per field (**e**; $p = 0.0358$). Accordingly, the average size per mitochondrion drops (**f**; $p = 0.0037$), although mitochondria still make up the same percentage of the area of the total field (**g**). **h** Mitochondrial DNA is reduced in hearts expressing IGF2BP2, as measured by PCR ($p = 0.0466$). **b** and **c** are aligned dot plots showing the mean value; **e**–**h** are scatter plots/bar graphs showing the mean and standard error of the mean. *$p < 0.05$, **$p < 0.01$ and ****$p < 0.001$.

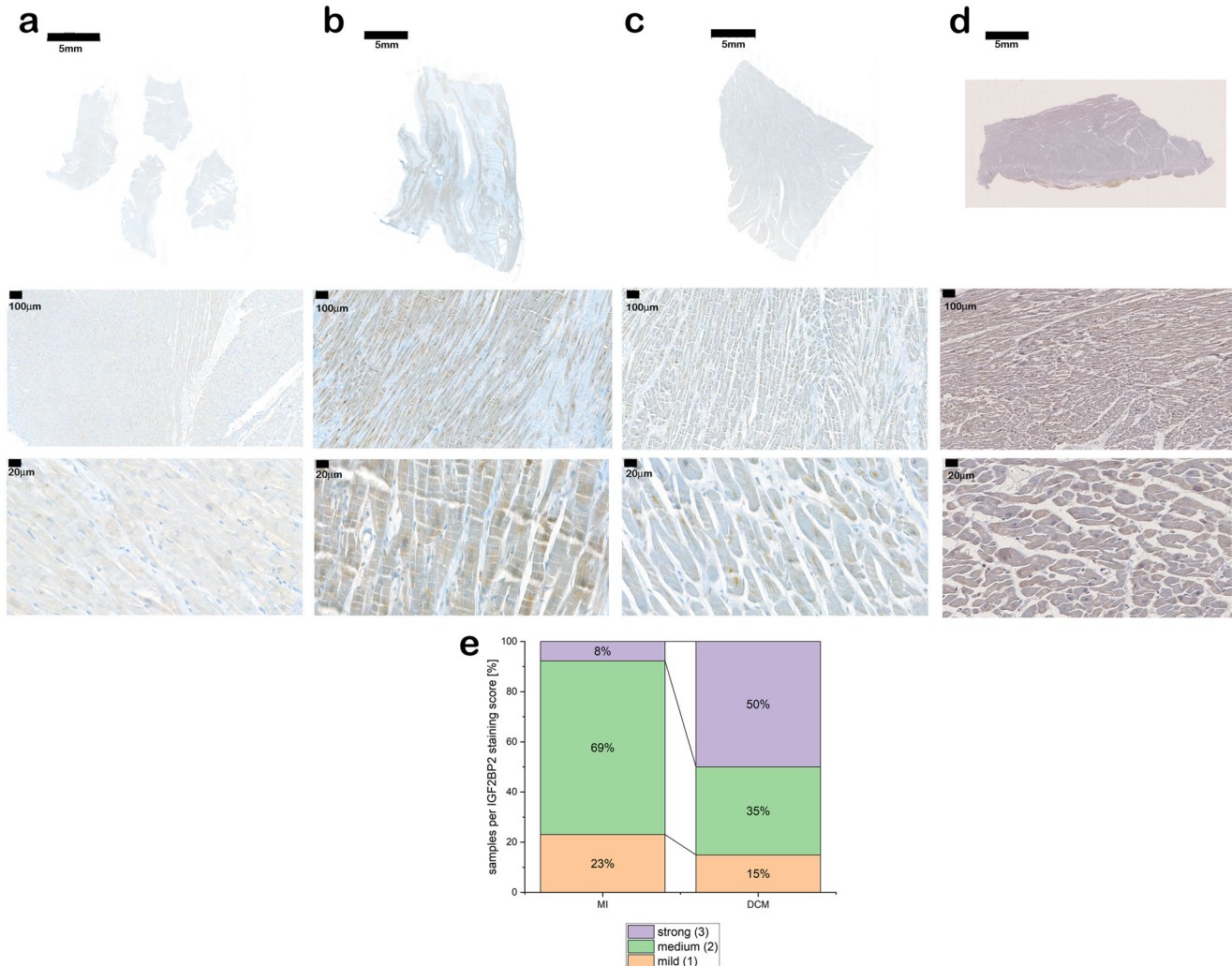

**Fig. 7 Expression of IGF2BP2 in human cardiomyopathies.** Human pathological heart sections were stained with antibody against IGF2BP2.
**a** Hypertrophic phenotype. **b** Dilatative cardiomyopathy. **c** Myocardial infarction. **d** Control heart. First row shows overview images at ×0.3 magnification, scale bar 5000 μm, the second row shows images at ×8 magnification, scale bar 100 μm, and the third row shows detail images at ×40 magnification, scale bar 20 μm. **e** The degree of IGF2BP2 expression was quantified as described in the "Methods" section. MI myocardial infarction, $n = 13$; and DCM, $n = 20$, $p = 0.0404$.

(Supplementary Data 1) and the mitochondria are fragmented (Fig. 6). While it is tempting to speculate that IGF2BP2 is repressing translation of many of the RNAs bound by the protein, we find that a number of the RNAs encoding downregulated mitochondrial (including OPA1) proteins are in fact downregulated at the RNA level. These results suggest that IGF2BP2 may be acting through multiple regulatory pathways to affect the adaptive response we observe. A systemic, multi-omics approach should help us better understand how IGF2BP2 coordinates the DCM phenotype. Integrating the proteomic analysis presented here with eCLIP analysis that identifies the direct targets of

IGF2BP2 and RNAseq data would help identify the compound mechanisms of IGF2BP2 action in hearts undergoing cardiac stress.

The proteomic data show that many sarcomeric proteins are also reduced by elevated IGF2BP2 expression. Inasmuch as some of the downregulated proteins showed no change in their RNA levels while others were reduced, IGF2BP2 may be working through different modes of action. Many of the downregulated sarcomeric or Z-disc-associated proteins were previously identified as genes associated with DCM in patients. Consistent with these findings, primary cardiomyocytes overexpressing

IGF2BP2 show a loss of directionality and polarization. At the EM level, the sarcomeres of hearts overexpressing IGF2BP2 are elongated and thinner, a phenotype observed in DCM hearts. Taken together, these data suggest that IGF2BP2 upregulation impairs the expression of many IGF2BP2 target RNAs, particularly in mitochondria and sarcomeres, leading to DCM. Future studies should help clarify the mechanisms responsible for the observed phenotypes.

In recent years, RNA methylation has been observed to play a role in cardiomyopathies. m6A RNA methylation is upregulated when heart failure occurs in both humans and mice, as well as in primary cultured cardiomyocytes exposed to ischemia[43]. When RNA methylation is upregulated either by overexpression of the methylation writer, METTL3A, or downregulation of the methylation eraser, FTO, hearts become hypertrophic[44] and show impaired cardiac contractility[43]. Conversely, mice in which METTL3A is knocked down do not become hypertrophic upon aging or when exposed to cardiac stress[44], and MI-induced cardiac injury can be rescued by overexpression of FTO[43]. The IGF2BP proteins have been proposed to be readers of m6A RNA methylation, i.e., methylation of their RNA targets enhances binding by the proteins[4]. It is tempting to speculate that increased m6A RNA methylation could lead to elevated RNA binding by IGF2BP2, facilitating its role in mediating cardiomyopathies. In this regard, it is interesting to note that *METTL3* RNA has been reported to be itself a target of IGF2BP2 in HEK293 cells[4]. In addition, IGF2BP2 has recently been shown to cooperate with miR-133a and AGO2 in an RNA methylation-dependent manner: increased m6A methylation leads to increased IGF2BP2 binding of *miR-133a* and recruitment of AGO2 to target RNAs, causing translational repression and cardiac hypertrophy[45]. Future experiments will more fully determine how IGF2BP2 may interact with the methylation machinery in the heart.

In summary, we propose a model in which cardiac stress upregulates IGF2BP2, which in turn induces cardiac remodeling that acts as an adaptive response, allowing the heart to continue to function. This activation causes a shift in the heart from mitochondrial- to non-mitochondrial-based metabolism, as well as a disruption of normal sarcomeric organization. The resulting dilated cardiomyopathy eventually leads to heart fatigue and death, unless IGF2BP2 expression is reduced in time. We propose that IGF2BP2 may be a useful target for the treatment of cardiac disorders in which there are upregulated levels of the protein.

## Methods

### Mice

*Transgenic animals*. All animal procedures were approved and performed in accordance with the local animal ethics committee (MD-17-15389, MD-20-16134).

TRE-human p62 (IGF2BP2) mice have been previously described[46]. In short, the human IGF2BP2 protein (p62) transgene is under the control of the trans-responsive element cytomegalovirus (TRE-CMVmin) promotor. In order to induce human *IGF2BP2* expression in the heart, transgenic mice were bred with αMHC-tTA mice (Jackson Labs), which carry the gene for tetracycline transactivator (tTA) under a cardiac-specific promoter (αMHC). Cardiac-specific expression of the transgene can be switched off by the application of tetracycline or switched on by the withdrawal of tetracycline. To repress human *IGF2BP2* expression during embryonic and early postnatal heart development, mothers were given 0.5 mg/ml tetracycline (Tamar Inc., Israel) with 3% sucrose in the drinking water. Mice were then kept on tetracycline until induction at 8-10 weeks of age. All heart weight to body weight (HW/BW) or tibia length to body weight (TL/BW) ratios were done per sex. Male mice were used for

IGF2BP2 rescue experiments, echocardiograms, and electron microscopy analyses. Female mice were used for proteomics and histology.

**Transverse aortic constriction (TAC)**. TAC mice models ($n = 3$) to induce chronic pressure overload of the left ventricle[20,47] were performed as described[48]. The procedure is described in detail on the following site https://app.jove.com/t/1729, but we used a 27 G needle (and not 27.5 G). The RNA was extracted at 10 weeks post-TAC, as described[48].

**Echocardiography**. Short axis measurements were obtained for heart function evaluation, by transthoracic echocardio-graphy performed on isoflurane-sedated mice using a Vevo LAZR-X VisualSonics device for IGF2BP2 mice and the VEVO700 for the IGF2BP1 mice. Three frames of short axis were analyzed for three diastole and systole cycles. All echocardiography measurements were performed in a blinded manner.

Data is presented as mean ± SEM. **$p < 0.01$, ****$p < 0.0001$; statistical significance was calculated using one-way ANOVA followed by Dunnett's post hoc test relative to tTA group.

**Histology staining and IHC**. Hearts were fixed and paraffin-embedded. Sections of 5 μm thickness were then processed for tissue staining. H&E and Masson Trichrome staining (for fibrosis) were done using standard protocol. Quantification of fibrosis was performed using FiJi software for creating masks. Immuno-histochemistry was performed using the primary antibodies summarized in Supplementary Table 1.

Autofluorescence in immunofluorescence staining was quenched using the Vector® TrueVIEW® Autofluorescence Quenching Kit (Vector Laboratories, SP-8400-15) following the manufacturer's instructions. Wheat germ-agglutinin staining (1 mg/ml, Sigma) was performed for 1 h, treated with a Vector TrueVIEW kit, stained with DAPI, and mounted with VECTA-SHIELD Vibrance Antifade Mounting Medium[44]. In enzymatic reactions, the ImmPRESS® HRP Horse Anti-Rabbit IgG Polymer Detection Kit, Peroxidase (Vector Laboratories, MP-7401) and the ImmPACT® DAB Substrate, Peroxidase (HRP) (Vector Laboratories, SK-4105) were used.

Hearts were cut in half in adults, taking the apex for molecular analysis and a region from under the midline to the base for sectioning. Sections prepared for experiments of IGF2BP2 expression from birth were prepared from the apex to the base region.

**Electron microscopy**. For perfusion, mice were weighed and anesthetized by i.p. injections with ketamine/xylazine solution (50 mg/kg ketamine/7.5 mg/kg xylazine in 0.9% NaCl solution). The animals were transcardially pre-perfused with ice-cold 10 ml of 0.1 M phosphate buffer, followed by perfusion with 10 ml ice-cold fresh prepared Karnovsky fixative containing 2% paraformaldehyde and 2.5% glutaraldehyde EM grade, in 0.1 M sodium cacodylate buffer pH 7.3.

Tissue was washed four times with sodium cacodylate and postfixed for 1 h with 1% osmium tetroxide and 1.5% potassium ferricyanide in sodium cacodylate, and washed 4 times with the same buffer, followed by dehydration with a graded ethanol series, followed by 2 changes of propylene oxide. Cells were then infiltrated with a series of epoxy resin and polymerized in the oven at 60 °C for 48 h. 80 nm sections were obtained and stained with uranyl acetate and lead citrate. Sections were observed by Jeol JEM 1400 Plus transmission electron microscope, and pictures were taken using a Gatan Orius CCD camera.

**Human sections**. Immunohistochemical staining was performed with the Ventana Benchmark Ultra System, using the Ultra View Universal DAB Detection Kit (#5269806001, Roche Diagnostics, Rotkreuz, Switzerland) for antibody detection, according to the manufacturer's instructions. Anti-IGF2BP2/p62 antibody[49], which detects both isoforms, was incubated in a 1:100 dilution for 32 min at room temperature. Epitope retrieval was achieved with Cell Conditioning Solution 1 (#5279801001, Roche Diagnostics, Rotkreuz, Switzerland) at 95 °C for 64 min. Samples were examined by an independent investigator blinded to experimental conditions. The intensity of each section was scored (intensity 0 = no detectable staining, 1 = weak staining, 2 = moderate staining, and 3 = strong staining). Statistics were done using the $\chi^2$ test.

Samples used in this project were obtained under informed consent at the Medical University of Innsbruck with the permission of the local ethics commission (EK number 4077).

**Cardiomyocyte isolation and immunofluorescence staining**. Cardiomyocytes were isolated using the Neonatal Cardiomyocyte Isolation System (#LK003300; Worthington Biochemical Corporation) following the manufacturer's instructions. Cardiomyocytes were seeded with DMEM-F12 (Biological Industries, Israel) with 10% FCS and 1% ciprofloxacin (Bayer) for 48 h and then grown in DMEM-F12 with 0.1% serum and 1% ciprofloxacin for 3–4 days on cover glasses coated with BME (Celtrux). Cells were fixed with 4% formaldehyde and incubated with primary and secondary antibodies consecutively (see the subsection "Histology staining and IHC" for a full list of antibodies).

Cells were imaged using the Nikon Spinning Disk microscope. For the directionality of pixels in MF20 cardiomyocyte staining, ImageJ software was used, using the OrientationJ plugin as instructed[28]. Three repeats of the experiment were performed.

**Sample preparation for MS analysis**. Heart tissue was homogenized in RIPA buffer containing protease and phosphatase inhibitors, clarified by centrifugation, and the supernatant was subjected to protein precipitation by the chloroform/ methanol method[50]. The precipitated proteins were solubilized in 100 µl of 8 M urea, 10 mM DTT, 25 mM Tris–HCl pH 8.0 and incubated for 30 min at 22 °C. Iodoacetamide (55 mM) was added followed by incubation for 30 min (22 °C, in the dark), followed by the re-addition of DTT (10 mM). 25 µg of protein was transferred into a new tube, diluted by the addition of 7 volumes of 25 mM Tris–HCl pH 8.0, and sequencing-grade modified Trypsin (Promega Corp., Madison, WI, USA) was added (0.4 µg/sample) followed by incubation overnight at 37 °C with agitation. The samples were acidified by the addition of 0.2% formic acid and desalted on C18 home-made Stage tips. Peptide concentration was determined by Absorbance at 280 nm and 0.75 µg of peptides were injected into the mass spectrometer.

**nanoLC–MS/MS analysis**. MS analysis was performed using a Q Exactive-HF mass spectrometer (Thermo Fisher Scientific, Waltham, MA, USA) coupled on-line to a nanoflow UHPLC instrument, Ultimate 3000 Dionex (Thermo Fisher Scientific, Waltham, MA, USA). Peptides dissolved in 0.1% formic acid were separated without a trap column over a 120 min acetonitrile gradient run at a flow rate of 0.3 µl/min on a reverse phase 25-cm-long C18 column (75 µm ID, 2 µm, 100 Å, Thermo Pep-MapRSLC). The instrument settings were as described[51]. Survey scans (300–1650 $m/z$, target value 3E6 charges, maximum ion injection time 20 ms) were acquired and followed by higher energy collisional dissociation (HCD) based fragmentation (normalized collision energy 27). A resolution of 60,000 was used for survey scans and up to 15 dynamically chosen most abundant precursor ions, with peptide preferable profiles, were fragmented (isolation window 1.8 $m/z$). The MS/MS scans were acquired at a resolution of 15,000 (target value 1E5 charges, maximum ion injection times 25 ms). Dynamic exclusion was 20 s. Data were acquired using Xcalibur software (Thermo Scientific). To avoid a carryover and to equilibrate the C18 column, the column was washed with 80% acetonitrile, and 0.1% formic acid for 25 min between samples, as per standard protocol.

Raw data were processed using MaxQuant (MQ) version 1.6.5.0[52] and the embedded Andromeda search engine[53]. The bioinformatics was performed with the Perseus suite (version 1.6.2.3). The data were filtered for reverse, contaminants and identified by site. Then the data were filtered such that a protein had to have non-zero LFQ intensity in all 9 samples with 3 or more peptides. The significantly enriched proteins were found (two-sample $t$-test with a permutation-based FDR method) and further selected using an adjusted $p$-value < 0.05 and S = 0.1. Log2-transformed individual values or triplicate means were $z$-score-normalized prior to hierarchical clustering. Gene Ontology annotation was performed using STRING site version 10.5, and the statistical values by the site have been reported. A file containing lists of the proteins at the different stages of filtering has been included as Expanded View data (Supplementary Data 1).

**Statistics and reproducibility**. Data are shown as mean ± SEM unless otherwise stated. One-way analysis of variance (ANOVA) was used to determine statistical significance for experiments with more than two groups followed by Dunnett's post hoc test. Figures with ANOVA analysis where applicable are indicated in corresponding figure legends. Comparison between the two groups was carried out using GraphPad software with an unpaired Student's $t$-test. In the mitochondrial and metabolic proteome comparison and in the human IHC comparison, statistics were performed using a chi-squared test. $p$-values < 0.05 were considered statistically significant and assigned in individual figures. Sample sizes and replicates are indicated in the legends for each experiment.

**Real-time and quantitative polymerase chain reaction**. Total RNA from heart tissue was extracted with the NucleoSpin RNA Mini Kit with DNase (Macherey-Nagel). cDNA was synthesized using oligo dT with the Flex Script cDNA Synthesis kit (Quanta Bio). qPCR was done on a CFX384 Touch™ Real-Time PCR Detection System (Bio-Rad) using the Fast SYBR Green Master Mix (Applied Biosystems, ThermoFisher Scientific). Results were calculated using Bio-Rad CFX software using two reference genes (Rpl4 and HPRT). The list of primers used is shown in Supplementary Table 2.

For nuclear and mitochondrial DNA extraction, the Qiagen DNeasy Blood & Tissue Kit (69504) was used. For nuclear DNA, a sequence on chromosome 12 was used as an internal control, and a sequence from COX1 was used for mitochondrial DNA.

**Western blot analysis**. Tissue extracts were prepared using RIPA buffer with the addition of (β-glycerophosphate and sodium orthovanadate) and a protease inhibitor cocktail (Ape Bio). 20 µg of protein extract were run per well on precast SDS–PAGE gels (Bio-Rad) and transferred to nitrocellulose membranes prior to imaging with the Bio-Rad ChemiDoc Imaging System. A list of relevant antibodies can be found in Supplementary Table 1.

**Reporting summary**. Further information on research design is available in the Nature Portfolio Reporting Summary linked to this article.

## Data availability

Uncropped and unedited blot/gel images are included in Supplementary Fig. 7. The source data behind the graphs in the paper can be found in Supplementary Data 2. The mass spectrometry proteomics data have been deposited in the ProteomeXchange Consortium via the PRIDE[54] partner repository with the dataset identifier PXD037623. Publicly available Gene omnibus (GEO) data sets from murine myocardial infarction samples from mice that underwent ligation of the left coronary artery (GDS488), heart failure due to expression of mutated estrogen receptor-coupled Cre under transcriptional control of cardiac-specific myosin heavy chain (MerCreMer; GDS5469), and induced cardiomyopathy due to exposure to isoproterenol (GDS3596), were analyzed for IGF2BP2 expression.

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

## Acknowledgements

Thanks to Dr. Jonathan Balkin for helpful comments, Rinat Abromovitz and all of the Wohl center staff, David Mimon and Tali Benkin, for assistance with the echocardiograms, Nadav Wallis and Hagar Klein for help with mice and cardiomyocyte cultures, Yoav Smith for assistance with the bioinformatics analysis, and Omri Yosef from Micha Berger's lab for the primer sequences for mitochondrial DNA. This study was funded by the Deutsche Forschungsgemeinschaft (#KE 2519 to J.K.Y. and S.M.K.), the American Friends of the Hebrew University, the Salomon Family Philanthropic Fund of the Hebrew University, and the Saul and Joyce Brandman Cardiovascular Research Hub of the Institute for Medical Research—Israel-Canada (IMRIC) at the Faculty of Medicine of The Hebrew University of Jerusalem.

## Author contributions

Project conception and planning: M.K., F.O., Y.C., R.B., S.M.K., and J.K.Y.; experiments: M.K., F.O., Y.C., M.G., V.B., I.F., J.H., and I.K.; analysis of data: M.K., D.M., G.V., V.B., K.M., I.F., J.H., G.P., S.M.K., and J.K.Y.; writing and editing of manuscript: M.K., F.O., S.M.K., and J.K.Y.

## Funding

## Competing interests

The authors declare no competing interests.
