## [Peer Review File · Communications Biology]

Reviewers' comments:

Reviewer #1 (Remarks to the Author):

In the current study, Krumbein et al. examined the effect of IGF2BP2 on the development of dilated cardiomyopathy. Using an in vivo Tet-off system, the researchers demonstrated that sustained cardiac expression of human IGF2BP2 for a period of 3-4 weeks resulted in dilated cardiomyopathy-mediated mortality, both in the postnatal and adult stages. Interestingly, induction of human IGF2BP2 for a shorter period of 14 days did not induce cardiac dysfunction or mortality. However, extending the induction of IGF2BP2 to 16 days resulted in severe impairment of cardiac function and morphology, with 2 out of 5 mice dying during the follow-up period. In addition, the authors observed an upregulation of IGF2BP2 protein levels in human patients having dilated cardiomyopathy. To gain mechanistic insight, the authors performed proteomic analysis, which revealed dysregulation of proteins associated with mitochondrial function in the IGF2BP2-expressing mouse heart.

Overall, these results clearly demonstrated that a prolonged expression of human IGF2BP2 can trigger the onset of cardiomyopathy in mice.

Major Comments:

1. The authors performed a proteomic analysis to investigate the mechanism by which exogenous expression of human IGF2BP2 induces cardiomyopathy. However, given that IGF2BP2 functions as a reader for m6A mRNA modifications, the primary target of IGF2BP2 should be the mRNA. It is likely that overexpression of hIGF2BP2 perturbs the transcriptome by altering the stability or localization of m6A-containing mRNAs that interact with IGF2BP2, ultimately leading to the translational defects. To elucidate the mechanism, the authors need to perform m6A RNAseq and/or conventional RNAseq in both control and hIGF2BP2-expressing hearts. Such experiments will provide valuable insights into the molecular mechanisms underlying the development of cardiomyopathy induced by IGF2BP2 overexpression.

2. Fig. 2F-Fig. 4: The observation that induction of hIGF2BP2 expression beyond 16 days, but not 14 days, leads to mortality and cardiac dysfunction suggests that the duration of hIGF2BP2 protein induction is critical for myocardial fate. However, the temporal dynamics of hIGF2BP2 protein have not been studied in detail. To better understand this phenomenon, the authors should assess the protein levels of huIGF2BP2 at different time points. In particular, it is crucial to determine the time point at which hIGF2BP2 protein becomes detectable after transgene induction and the time point at which hIGF2BP2 disappears from the hearts after transgene elimination.

Minor comments:

3. Fig. 1C: It is not clear whether the authors generated the sFlt-expressing mice and obtained the expression data from their mice or reanalyzed the expression data from the previous study. If the former is the case, the authors need to add a description of the transgenic mice in the Methods section. If the latter is the case, Figure 1c should be moved to Supplementary Figure 1.

4. Fig. 1E-F: The experimental condition of Figure 1E-F is not clear. Because the expression level of IGF2BP2 seems to peak 24~48 h after TAC surgery and decrease to the basal level (Supplementary Figure 1A), authors need to specify the time point related to the data of Fig. 1E-F. In addition, authors need to describe the TAC procedure in detail in the Methods section. For Figure 1F, DAPI and IGF2BP2 images should be shown separately.

5. line 184-185, related to Fig. 4: To claim significance, the authors must perform a Kaplan-Meier test and report the P value.

6. Supplemental fig 5: No scale bars in the images.

Reviewer #2 (Remarks to the Author):

The authors describe the role of IGF2BP2 in early-stage cardiac stress. This elegant study includes mouse models, primary cell cultures, and human patient material all confirming their findings. The study paves the path for novel studies into the potential therapeutic implications of IGF2BP2.

Please remove the double spaces throughout the manuscript (lines 218, 223, 227, 250, and perhaps more)

Line 86-98 describes the expression of IGF2BP2 in cardiac stress models in literature and therefore seems to be introduction or discussion, more than something for the results section

Line 122: please remove the remark "(see Materials and Methods)"

Line 135: please explain what is meant by "sparse"

Line 141: please refer to Figure 2G

Line 192: minimal fibrosis is described in mouse V, however on the picture it seems to have increased fibrosis more on the epicardial side of the heart, which would be different from the other hearts.

Please describe if this is the case.

Line 231: MF20 was stained in the primary cardiomyocytes, however, it would be better to stain for any of the sarcomeric proteins like α Actinin (line 246) which is down-regulated (antibody available according to the manuscript) and also one of the upregulated proteins

Lines 259-270: This part rather seems to be a discussion instead of a result section.

Line 276: the text indicates that multiple sections of human HCM patients have been used, however, it is not indicated how many samples have been used. Also, the qualitative scoring should be included in the figure.

Line 415: dilution missing for α Actinin Western blot

Line 525: Unfortunately, just one gene (HPRT) is used as a reference gene for normalization. Why not perform proper normalization by measuring multiple refgenes and define the optimal number by genorm

Figure 1:

This figure is messy due to several reasons:

- 1) The bar graphs are depicted in different styles
 - a. B is entirely different from the rest
 - b. A and C have a title whereas E does not
- 2) The ChIP (please check the legend how it should be written) data is barely visible due to low quality
- 3) The IF staining should show a view on a higher magnification
- 4) The bar graph should include the dot plots as shown in 1N, 1O, 1P, 2, and 6

Figure 2:

This figure includes lots of relevant information, but the layout is messy: Figure panels have different sizes (echocardiogram huge compared to the rest), font sizes differ, and therefore most of the text is difficult to read, lots of white spots in the figure. The fibrosis content in Masson`s Trichrome scan could be quantified. I don't see additive value in panels B and F. The legend indicates the precise p-

values between brackets, but p-values are also indicated in asterisks. Please remove the precise numbers from the legend.

Figure 7:

For comparison, this Figure would be more illustrative when a healthy heart would be included.

Table 1:

Please add that you are looking at protein expression in the legend

Reviewer #3 (Remarks to the Author):

The manuscript by Krumbein et al investigates the effect of IGFBP2 over-expression in mouse hearts using a conditional tg model. The authors show that IGFBP2 over-expression is detrimental and that this response is time-dependent. The authors further show that the several proteins are dysregulated in these animals, and that these proteins are mostly mitochondrial or sarcomeric.

The main issue with the paper is the conclusion that IGFBP2 has a direct impact on translation of these mRNAs. The observed changes can be the consequence of pathologic remodeling and not a direct consequence of IGFBP2 over-expression. Immunoprecipitation would be needed to make these statements. This reviewer suggests adding this to the limitations and to limit these conclusions in the text.

In Figure 1B, the levels of IGF2BP2 seem highly variable in P1. The quantification of IGF2BP1 is higher than BP2, but that is not a reflection of the blots. A better blot is needed.

The variability in the response after 16 days of transgene expression is surprising. What is the sex of those animals?

The authors state that several proteins are evaluated by Western blot, whereas only 4 are evaluated. Only one is a mitochondrial protein. Proteomics is sufficient for the protein analysis, but the authors should minimize the relative importance of the Western data in the text.

The authors can speculate on the effect of IGFBP2 on RNA stability in the Discussion, but not in the results, as they have no evidence that it is increasing RNA stability.

Quantification in Figure 7 needs to be shown as a bar graph.

This reviewer agrees that timing of IGFBP2 regulation and suppression are likely important for a potential recovery. The authors need to show a time-course of IGFBP2 increased levels and the relationship to TAC and sFLT up-regulation and the association between its levels and cardiac function.

Line 328 – the authors state that they observed association of IGFBP2 with mitochondrial RNAs. The authors don't provide any association data. Do they mean association in the levels of IGFBP2 and RNAs? As written, this is misleading. In addition, I suspect these are not coded by the mitochondrial genome, correct? Please clarify if any of the identified proteins are coded by the mitochondria.

Dear Referees,

Thank you for your comments and recommendations. We have addressed all of the points mentioned in the reports. Each point is listed below, with our responses highlighted in yellow. Figures 1,2,5, and 7 and Extended Data Fig. 6 have been changed according to your requests, and new data has been added in Extended Data Figures 2A, B, F, and 5. We hope that you will agree with us that the paper has been significantly improved and is now appropriate for publication in Communications Biology.

Sincerely,

Joel Yisraeli

Reviewer #1 (Remarks to the Author):

Major Comments:

1. The authors performed a proteomic analysis to investigate the mechanism by which exogenous expression of human IGF2BP2 induces cardiomyopathy. However, given that IGF2BP2 functions as a reader for m6A mRNA modifications, the primary target of IGF2BP2 should be the mRNA. It is likely that overexpression of hIGF2BP2 perturbs the transcriptome by altering the stability or localization of m6A-containing mRNAs that interact with IGF2BP2, ultimately leading to the translational defects. To elucidate the mechanism, the authors need to perform m6A RNAseq and/or conventional RNAseq in both control and hIGF2BP2-expressing hearts. Such experiments will provide valuable insights into the molecular mechanisms underlying the development of cardiomyopathy induced by IGF2BP2 overexpression.

We certainly agree that the question of the mechanism by which IGF2BP2 is working is very interesting. As the editors indicated, we have chosen to take a targeted approach to address the reviewer's comment. We have now performed rtPCR on 7 different genes that were downregulated in the proteomics data. The results, presented in Extended Data Fig. 5, show that some RNAs are downregulated, while others are not. We suspect this indicates that IGF2BP2 may be working through different regulatory pathways. The different possibilities are discussed in the Discussion. (Lines 231-238 and 331-345)

2. Fig. 2F-Fig. 4: The observation that induction of hIGF2BP2 expression beyond 16 days, but not 14 days, leads to mortality and cardiac dysfunction suggests that the duration of hIGF2BP2 protein induction is critical for myocardial fate. However, the temporal dynamics of hIGF2BP2 protein have not been studied in detail. To better understand this phenomenon, the authors should assess the protein levels of hIGF2BP2 at different time points. In particular, it is crucial to determine the time point at which hIGF2BP2 protein becomes detectable after transgene induction and the time point at which hIGF2BP2 disappears from the hearts after transgene elimination.

The timing of the induction and silencing of IGF2BP2 expression in the transgenic mice indeed bears on the question of the rescue experiment. We have added a figure showing the kinetics of IGF2BP2 expression following withdrawal or addition of Tet to the drinking water. These data are shown in Extended Data Fig. 2A and B.

Minor comments:

3. Fig. 1C: It is not clear whether the authors generated the sFlt-expressing mice and obtained the expression data from their mice or reanalyzed the expression data from the previous study. If the former is the case, the authors need to add a description of the transgenic mice in the Methods section. If the latter is the case, Figure 1c should be moved to Supplementary Figure 1.

We apologize for the lack of clarity. We did in fact reanalyze data from a previous study. The figure now appears in Extended Data Fig. 1D and E.

4. Fig. 1E-F: The experimental condition of Figure 1E-F is not clear. Because the expression level of IGF2BP2 seems to peak 24~48 h after TAC surgery and decrease to the basal level (Supplementary Figure 1A), authors need to specify the time point related to the data of Fig. 1E-F. In addition, authors need to describe the TAC procedure in detail in the Methods section. For Figure 1F, DAPI and IGF2BP2 images should be shown separately.

The details of the TAC surgery have been added to the Materials and Methods section (lines 401-403) along with the appropriate references. The RNA samples were taken 10 weeks after the surgery, and this is now explicitly mentioned in the legend and the M&M description. We apologize for any misunderstanding regarding the protocol.

5. line 184-185, related to Fig. 4: To claim significance, the authors must perform a Kaplan-Meier test and report

the P value.

The text has been changed to "...the outcome in this group was variable" (line 184)

6. Supplemental fig 5: No scale bars in the images.

We have added a scale bar to the figure (which is now Extended Data Fig. 6).

Reviewer #2 (Remarks to the Author):

The authors describe the role of IGF2BP2 in early-stage cardiac stress. This elegant study includes mouse models, primary cell cultures, and human patient material all confirming their findings. The study paves the path for novel studies into the potential therapeutic implications of IGF2BP2.

Please remove the double spaces throughout the manuscript (lines 218, 223, 227, 250, and perhaps more)

I have tried to remove them, but in some cases it was a result of the Word justification program. In addition, I was taught that after a period, one puts in a double space before beginning the next sentence.

Line 86-98 describes the expression of IGF2BP2 in cardiac stress models in literature and therefore seems to be introduction or discussion, more than something for the results section

Because this discussion includes new analysis of previously published data (Extended Data Fig. 1A,B,and C, we felt it was more appropriate in the results section.

Line 122: please remove the remark "(see Materials and Methods)"

removed

Line 135: please explain what is meant by "sparse"

Changed to "less dense" (line 134)

Line 141: please refer to Figure 2G

The figure reference has been added (it is now Fig. 2E).

Line 192: minimal fibrosis is described in mouse V, however on the picture it seems to have increased fibrosis more on the epicardial side of the heart, which would be different from the other hearts. Please describe if this is the case.

The pathologist who examined the sections was not certain that the slight differences to which the reviewer refers were significant or perhaps just a result of fixation/sectioning/staining.

Line 231: MF20 was stained in the primary cardiomyocytes, however, it would be better to stain for any of the sarcomeric proteins like α Actinin (line 246) which is down-regulated (antibody available according to the manuscript) and also one of the upregulated proteins

The reviewer raises an interesting question about whether we could see in primary cardiomyocytes the upregulation or downregulation of the proteins that we observed in the induced hearts. We were interested, however, in simply looking to see if the organization of the sarcomere was affected. The question of how well the primary cardiomyocytes reflect what is happening in the whole heart is a question we hope to address in the future.

Lines 259-270: This part rather seems to be a discussion instead of a result section.

The section has been slightly changed to reflect the addition of new data (OPA1 rtPCR). We felt that it was important to give a bit of context for the experiments. (lines 267-269)

Line 276: the text indicates that multiple sections of human HCM patients have been used, however, it is not indicated how many samples have been used. Also, the qualitative scoring should be included in the figure.

The number of samples is now indicated in the legend, and the scoring is now shown as a bar graph.

Line 415: dilution missing for α Actinin Western blot

In order to streamline the paper a bit, this western blot has been removed.

Line 525: Unfortunately, just one gene (HPRT) is used as a reference gene for normalization. Why not perform proper normalization by measuring multiple refgenes and define the optimal number by genrom

We have done new rtPCRs using two reference genes and analyzed accordingly. These are shown in Extended Data Fig. 5 and primers are listed in Materials and Methods.

Figure 1:

This figure is messy due to several reasons:

- 1) The bar graphs are depicted in different styles
 - a. B is entirely different from the rest
 - b. A and C have a title whereas E does not
- 2) The ChIP (please check the legend how it should be written) data is barely visible due to low quality
- 3) The IF staining should show a view on a higher magnification
- 4) The bar graph should include the dot plots as shown in 1N, 1O, 1P, 2, and 6

The figure has been reformatted and reorganized along the lines that the reviewer indicated.

Figure 2:

This figure includes lots of relevant information, but the layout is messy: Figure panels have different sizes (echocardiogram huge compared to the rest), font sizes differ, and therefore most of the text is difficult to read, lots of white spots in the figure. The fibrosis content in Masson's Trichrome scan could be quantified. I don't see additive value in panels B and F. The legend indicates the precise p-values between brackets, but p-values are also indicated in asterisks. Please remove the precise numbers from the legend.

The figure and legend have been reformatted and reorganized along the lines that the reviewer indicated. Quantification of fibrosis is shown in Extended Data Fig 2F.

Figure 7:

For comparison, this Figure would be more illustrative when a healthy heart would be included.

A panel showing a healthy heart has been added.

Table 1:

Please add that you are looking at protein expression in the legend

Done

Reviewer #3 (Remarks to the Author):

The manuscript by Krumbein et al investigates the effect of IGFBP2 over-expression in mouse hearts using a conditional tg model. The authors show that IGFBP2 over-expression is detrimental and that this response is time-dependent. The authors further show that the several proteins are dysregulated in these animals, and that these proteins are mostly mitochondrial or sarcomeric.

The main issue with the paper is the conclusion that IGFBP2 has a direct impact on translation of these mRNAs. The observed changes can be the consequence of pathologic remodeling and not a direct consequence of IGFBP2 over-expression. Immunoprecipitation would be needed to make these statements. This reviewer suggests adding this to the limitations and to limit these conclusions in the text.

We agree completely with the reviewer about the limitations of conclusions about how IGFBP2 is functioning. In addition to adding new rtPCR experiments on a number of RNAs encoding downregulated proteins (Extended Data Fig. 5), we have rewritten the sections related to possible modes of action for IGFBP2, taken speculations out of the Results section, and put in the Discussion possibilities for how the protein may be functioning. (Lines 231-238 and 331-345)

In Figure 1B, the levels of IGFBP2 seem highly variable in P1. The quantification of IGFBP1 is higher than BP2, but that is not a reflection of the blots. A better blot is needed.

Because each antibody has its own affinity, it is not possible (with simple western blots) to compare the expression of the IGFBP paralogs to each other but only relatively to themselves, between time points. Indeed, the way we presented the data was confusing. We have now separated the data, presenting on each graph the normalized, relative amount of one of the paralogs.

The variability in the response after 16 days of transgene expression is surprising. What is the sex of those animals?

Only males were used for the rescue experiments. This is now stated explicitly in the Materials and Methods section. (Lines 396-398)

The authors state that several proteins are evaluated by Western blot, whereas only 4 are evaluated. Only one is a mitochondrial protein. Proteomics is sufficient for the protein analysis, but the authors should minimize the relative importance of the Western data in the text.

Agreed – the section has been rewritten. (Lines 206-207)

The authors can speculate on the effect of IGFBP2 on RNA stability in the Discussion, but not in the results, as they have no evidence that it is increasing RNA stability.

Agreed – see answer above

Quantification in Figure 7 needs to be shown as a bar graph.

Done

This reviewer agrees that timing of IGFBP2 regulation and suppression are likely important for a potential recovery. The authors need to show a time-course of IGFBP2 increased levels and the relationship to TAC and sFLT up-regulation and the association between its levels and cardiac function.

We have added a time course of up and down regulation of IGF2BP2 expression in the transgenic mice (Extended Data Fig. 2A,B).

Line 328 – the authors state that they observed association of IGFBP2 with mitochondrial RNAs. The authors don't provide any association data. Do they mean association in the levels of IGFBP2 and RNAs? As written, this is misleading. In addition, I suspect these are not coded by the mitochondrial genome, correct? Please clarify if any of the identified proteins are coded by the mitochondria.

We agree completely and apologize for the unclear text. The whole section has been rewritten to indicate the possible ways in which IGF2BP2 might regulate protein expression (directly, indirectly, RNA levels, RNA translation) but we leave it open for further studies. (Lines 331-345)

Here are the updated figures, with an explanation of what has been changed:

Fig. 1 - The bar graphs are now dot plots in the same style. The ChIP panel has been moved, along with the associated bar graph, to Extended Data Fig. 1. Panel D has been separated to show the different channels. The graphs in panel B have been separated to show each paralog separately.

H

10 Days

Fig. 2 (above). The timeline panels in the previous version have been removed. The panels have been reformatted, in line with the reviewer's comments.

Fig. 7 – A control heart section has been added (panel D). Quantification of the scoring has been added (panel E).

Extended Data Fig. 2 – Panels A and B show the modulation of IGF2BP2 expression in the transgenic mouse model. Panel F shows the quantification of the fibrosis seen in Fig. 2H.

Extended Data Fig. 5 – rtPCR analysis of target RNAs encoding proteins downregulated in the proteome analysis.

Extended Data Fig. 6 – a scale bar has been added to the figure.

REVIEWERS' COMMENTS:

Reviewer #1 (Remarks to the Author):

The authors have made commendable efforts to address several of my concerns; however, the most critical question remains unanswered: What is the molecular mechanism underlying IGFBP2-induced cardiac dysfunction? To gain a deeper understanding of this mechanism, I strongly recommend that the authors perform more comprehensive experiments, such as RNA-seq, m6A-seq, or CLIP, to explore the transcriptome landscape in conjunction with the proteomic data. This approach would help identify the direct targets of IGFBP2 and provide valuable molecular insights into pathogenesis.

Unfortunately, the authors chose to use quantitative PCR (qPCR) to assess the expression levels of only seven genes related to mitochondrial and cardiac functions. Their results indicate that five genes are downregulated, while two genes show no significant differences. The authors concluded that IGFBP2 may act through alternative regulatory pathways based on this limited set of qPCR experiments. This conclusion is not sufficiently supported and I am not convinced by this reasoning.

Once again, I would like to emphasize that one of the most important issues is the identification of the direct targets of IGFBP2. Authors must provide convincing evidence or data to address this critical issue.

Reviewer #2 (Remarks to the Author):

The authors have provided satisfactory responses to all the questions raised, making the manuscript suitable for publication.

Reviewer #3 (Remarks to the Author):

The authors have addressed my prior concerns

Response to reviewers

Reviewer #1

In the revised MS we submitted, we took a targeted approach to look at the effects of IGF2BP2 up regulation on RNA expression. Of the 7 genes we tested (whose protein levels were all down regulated in IGF2BP2-overexpression hearts), 5 were down regulated at the RNA transcript level and 2 did not change significantly. Our analysis of these results was that IGF2BP2 might act in different ways to ultimately regulate protein levels in DCM hearts, and this is what we explained briefly in the Results and more extensively in the Discussion. We absolutely agree with the reviewer that the question of how IGF2BP2 leads to the DCM response on the molecular level is a very important issue. We think that our, admittedly small, sample of genes nevertheless already suggests that the regulation may be complex and vary for different genes. To fully address this question, we really need a systematic, multiomics approach, comparing eCLIP, RNAseq, and proteomics data. In addition, RNAs may be regulated at multiple levels, and this would involve testing both stability and translational control. An explanation of these questions and how to approach them in the future has now been included in the Discussion, but these experiments are well beyond the scope of the current paper. Ultimately, of course, it is the protein levels that determine the phenotype, and this analysis has been done on a global level and is presented in the paper.